# The Potential Applications of Commercial Arbuscular Mycorrhizal Fungal Inoculants and Their Ecological Consequences

**DOI:** 10.3390/microorganisms10101897

**Published:** 2022-09-23

**Authors:** Sulaimon Basiru, Mohamed Hijri

**Affiliations:** 1African Genome Center, Mohammed VI Polytechnic University (UM6P), Lot 660, Hay Moulay Rachid, Ben Guerir 43150, Morocco; 2Institut de Recherche en Biologie Végétale, Département de Sciences Biologiques, Université de Montréal, 4101 Sherbrooke Est, Montréal, QC H1X 2B2, Canada

**Keywords:** arbuscular mycorrhizal fungi, symbiosis, commercial inoculants, community structure, ecosystem functions, metabarcoding

## Abstract

Arbuscular mycorrhizal fungal (AMF) inoculants are sustainable biological materials that can provide several benefits to plants, especially in disturbed agroecosystems and in the context of phytomanagement interventions. However, it is difficult to predict the effectiveness of AMF inoculants and their impacts on indigenous AMF communities under field conditions. In this review, we examined the literature on the possible outcomes following the introduction of AMF-based inoculants in the field, including their establishment in soil and plant roots, persistence, and effects on the indigenous AMF community. Most studies indicate that introduced AMF can persist in the target field from a few months to several years but with declining abundance (60%) or complete exclusion (30%). Further analysis shows that AMF inoculation exerts both positive and negative impacts on native AMF species, including suppression (33%), stimulation (38%), exclusion (19%), and neutral impacts (10% of examined cases). The factors influencing the ecological fates of AMF inoculants, such as the inherent properties of the inoculum, dosage and frequency of inoculation, and soil physical and biological factors, are further discussed. While it is important to monitor the success and downstream impacts of commercial inoculants in the field, the sampling method and the molecular tools employed to resolve and quantify AMF taxa need to be improved and standardized to eliminate bias towards certain AMF strains and reduce discrepancies among studies. Lastly, inoculant producers must focus on selecting strains with a higher chance of success in the field, and having little or negligible downstream impacts.

## 1. Introduction

Mitigating the environmental impacts of intensive agriculture, such as greenhouse gas emission, eutrophication, the pollution of surface and underground water, global soil loss to salinity and compaction, loss of microbial diversity, etc., calls for the integration of multiple sustainable management approaches including the diversification of crop rotations, intercropping, integrated farm management, conservation, precision agriculture, and the ‘4R’ framework of nutrient management (meaning applying the right type and quantity of nutrient in right place and at the right time). Another important strategy that can help to reduce the environmental footprint of agriculture, restore soil health, and protect biodiversity is the introduction of beneficial microorganisms into agroecosystems [1,2].

Arbuscular mycorrhizal fungi (AMFs) are essential soil microbial communities that form obligate symbiosis with 80% of terrestrial plants [3]. AMF promotes plant growth by facilitating nutrient acquisition through the extraradical mycelia that spread from the host’s roots into surrounding soils [4,5]. By increasing plant access to nutrients, AMF application can offset phosphorus fertilizer demand by ~50% [6]. A growing number of studies also indicate that the application of AMF can reduce nitrogen losses in the form of nitrous oxide emission and nitrate leaching [7,8,9,10]. AMF can also stimulate the production of phytohormones and secondary metabolites, that can improve plant productivity [11], crop quality [12,13], and build resilience against environmental stress such as salinity [14], drought [15,16], heat, and pathogens [17,18]. Thus, intervention with commercial AMF is tenable in soils with high P-fixing potential, anthropized sites that exhibit low diversity and richness of native AMF species, and arid and semi-arid areas [19,20,21,22,23,24]. 

The AMF inoculant market is burgeoning and gaining wide recognition, although commercial products are majorly based on a few AMF strains belonging to the *Glomeraceae* genera: *Rhizophagus*, *Funneliformis*, and *Claroideoglomus* strains [2]. AMF inoculants consisting of several AMF strains can produce additive or synergistic effects on target crops [25], however increasing the diversity of AMF in inoculants may fail to produce additional benefits if constituent strains are redundant i.e., performing similar functions [26]. Similarly, the concerted application of AMF with other bioinoculants such as growth-promoting rhizobacteria (PGPR) or nitrogen-fixing bacteria, and organic substances such as humic acid can produce greater plant response [27,28]; however, only a few commercial products are based on such complex formulation [2].

Despite the growing demand for biofertilizers, results from commercial AMF inoculants have been largely context-dependent, especially under field conditions, contrary to the common laboratory successes [29,30,31,32]. The inconsistent narrations from the research community regarding the reliability of AMF inoculation as a valid agricultural management technique [33] are generating low consumer confidence that is hindering large-scale adoption of the technology. Although, quantifiable or marketable gains (such as yield enhancement, nutrient replacement, watering costs, seedling growth, and nursery raising) are the mostly sought benefits of AMF, agricultural benefits of AMF extend beyond immediate monetary gains. The non-marketable benefits of AMF include the improvement of food and fiber quality; plant stress mitigation as well as ecosystem services, such as soil erosion control; the prevention of nutrient losses (nitrous oxide emission and N leaching); carbon sequestration; and landscape recreation [34,35]. Notwithstanding, to win growers’ confidence and market sustainability, investment in inoculant technology must pay off either in the short- or long-term. 

Field success of commercial inoculants relies on the ability of the introduced strains to establish and persist at the target site for a desired or specified period. However, AMF establishment is complex and are often determined by the inherent features of the target agroecosystem. Unfortunately, most microbial inoculants are selected based on their expressed functional traits in a greenhouse, without proper consideration of the ecologically relevant traits that determine establishment and persistence under natural conditions [36]. While many cases of inoculation failures could be attributed to poor product quality stemming from the lack of regulatory or quality control frameworks that mandate best practices, resulting in a market flooded by substandard products [29,32,37], there is also lack of understanding of ecology and of the mode of action of inoculants [38]. Duell et al. [29] recently demonstrated that inoculants could decouple native plant symbiosis with indigenous strains in a natural ecosystem already containing a large diversity of AMF community, without conferring additional benefits. Thus, inoculation in such ecosystems with high mycorrhizal potential may prove unproductive. 

Therefore, understanding the factors that influence the establishment and persistence of AMF inoculants will inform decisions, as well as the management practices necessary, to ensure inoculation success [39]. Tracing the survival and persistence of the introduced strains will help identify which AMF strains adapt well to the local conditions and induce the observed plant response following long-term persistence and will help determine whether another inoculation exercise is necessary. Moreover, monitoring the downstream impacts of the introduced inoculant [40] is necessary to prevent the intentional propagation of detrimental or ineffective strains and the elimination of keystone taxa [41].

In the past, efforts to keep track of inoculants in the field have been scarce, resulting in knowledge gaps about persistence of introduced AMF as well as the downstream pressure/disturbance imposed on the native community. In part, this could be attributed to a lack of the molecular tools necessary to distinguish between related taxa [42]. However, highly sensitive and robust molecular markers coupled with rapid and cost-effective sequencing technologies are becoming increasingly accessible to resolve AMF species diversity with greater accuracy and precision. Consequently, a growing number of studies are monitoring the fate of AMF inoculants in the field. Therefore, this paper reviews the possible outcomes following the introduction of AMF inoculants into field soils. We highlight the survival limit or persistence of introduced AMF inoculum based on available evidence, and we also examine what factors drive the various outcomes post-inoculation. Lastly, we identify the impacts of foreign AMF inoculants on indigenous AMF communities and discuss the need to harmonize the techniques employed in monitoring inoculants in the field. 

## 2. Analysis of Published Studies on the Impact of Introduced AMF Inoculants in the Field

Most studies investigating AMF inoculants are conducted in greenhouses or in microcosms, using sterilized soils to compare the performance of non-AMF- and AMF-inoculated plants. This approach does not only fail to consider the realities of the agricultural field or agroecosystem, it also fail to capture the interactions of inoculants with indigenous species [43]. Field studies on AMF are notably scarce [44], especially those investigating the ecological consequences of inoculation [42]. In December 2021, we performed search operation on Scopus and Web of Science to collect studies monitoring AMF inoculants in the field using metabarcoding techniques. We used the following keywords: *“persistence” or “survival” and “arbuscular mycorrhizal fung*”*, *“inocula*”* and we recovered 407 references ( After screening of titles and abstracts, ten studies (16 site-specific studies) that fulfilled the screening criteria that is, carried out in the field and used metabarcoding to detect AMF in plant roots or soil, were selected for further examination (Table 1). The search was updated in September 2022 using using the following keywords: (“*arbuscular mycorrhizal fung**”, “*inocula**” *and* “*resident or “indigenous” or “autochthonous” and “community*” and “*field condition*”). A total of 136 articles were retrieved from both databases involving studies conducted between 1980–2022. After screening the abstracts, 15 studies that investigated the impact of introduced AMF on indigenous AMF communities in the field using molecular tools were further examined, culminating in a total number of 23 independent (site-specific) studies (Appendix A, Table 2). A separate search was carried out on the same databases to screen studies that assessed the persistence of introduced AMF in the field using metabarcoding techniques. Although we recovered 407 articles based on the search inputs (persistence or survival and arbuscular mycorrhiza* and field), only 10 articles that fulfilled the screening criteria were further selected for further examination. We analyzed the studies to identify the location of experiment and the identity of AMF. All AMF investigated belong to *Rhizophagus* (46%), *Funneliformis* (41%), *Claroideoglomus* (15%), *Glomus* (9%) and *Gigaspora* (2%) (Figure 1), corresponding to the AMF strains frequently obtained in commercial inoculants as reported in our previous study [2]. Moreover, the studies were conducted in ten countries located in different continents: Europe (10), North America (5), Asia (4) South America (2) and Africa (1) (Appendix A).

Further examination of the studies indicate that introduced AMF can survive in the field up to four years post inoculation. This survival period was perhaps limited by the study duration, which peaked at 4 years (Table 1). However, most of the studies reported a time-dependent decline in abundance of the introduced strains (60%), while others observed complete exclusion (29%) or no change (14%) (Figure 1B). Concerning the effects of introduced AMF on the indigenous AMF communities, we identified varied impacts that include suppression, stimulation, exclusion, or neutral effects (Figure 2). Mixed effects were observed in same experimental locations where certain AMF strains were stimulated, at the same time, others were suppressed or excluded (Figure 2 and Table 2). Lastly, the studies employed different markers targeting specific DNA regions (usually the nuclear ribosomal DNA or mitochondrial rDNA gene) as well as different quantification and sequencing platforms (Table 2). 

## 3. Establishment and Persistence of Introduced AMF in Field

### 3.1. Establishment

The influence of AMF on plants and microbial communities relies both on the successful colonization of plant roots and on the maintenance of live propagules that can form mycorrhizal association for a given period after their application [64]. However, the establishment success of introduced AMF in the field is context-specific, varying according to the type and composition of the inoculant, as well as the physical and biotic conditions of the target field [31]. The establishment of AMF is largely successful regardless of soil type, nutrient concentration, or the composition of the indigenous AMF communities (Table 1). In other contexts, AMF may fail to establish at the target site due various factors. For example, *Rhizophagus irregularis* (DAOM197198), often regarded as a generalist AMF with a cosmopolitan distribution, did not influence grapevine growth in a five-year trail [65,66]. Renault et al., 2020 [57] also observed no difference in root colonization between control and non-inoculated plots of corn soybean, or in wheat treated with same isolate of *R. irregularis*. In composite trials, where AMF inoculants were applied in different fields, establishment was successful at one site and failed at the other [60,61].

Surprisingly, there seems to be greater discrepancies among commercial and laboratory inoculants tested in the same field. In 2017, Berruti et al. [53] reported that a commercial inoculant failed to establish in the field, while other studies obtained inconsistent results, where some commercial inoculants survived better under greenhouse conditions [30,31,32]. A recent global evaluation of commercial AMF inoculants in greenhouse and field conditions demonstrated that only 4 out of 28 AMF inoculants were successfully established in a greenhouse experiment, and only one successfully influenced plant performance in the field, whereas the inoculum obtained from the laboratory established successfully in both ecosystems [32].

### 3.2. Persistence

Most studies indicate that AMF could survive for a long period after the first inoculation. More specifically, introduced AMF could survive for relatively few months after inoculation to several years (Table 1); however, no study has monitored the persistence of AMF in the field beyond four years, according to the information available while gathering data for this study. The survival limit, as well as the abundance of the introduce species, is majorly affected by the type of AMF as well as the biotic and abiotic conditions of the target sites. For example, out of the two non-native *Funneliformis mosseae* isolates—AZ2256C and IMA1, which originated from the USA and the UK, respectively—only the former was detected two years after inoculation, but in lower proportions compared to the native *R. irregularis* and *R. intraradices*, despite early establishment success and the stimulatory effects of both strains on the roots of *Medicago sativa* [67]. In a recent study, Pellegrino et al. [45] also reported that although the AMF strains *F. mossaeae* BEG12 and AZ225C were detected in alfalfa roots two years post-treatment, the AZ225C strains had longer persistence. 

Does persisting inoculant consistently improve plant growth, as in the initial stage of its introduction? Results from a 2012 study by Pellegrino et al. [48] suggest that persistent AMF could sustain yields two years after inoculation. A recent study by Pellegrino et al. [45] also indicated that AMF could sustain positive effects on the host crop several months after inoculation, whereby both *F. mossaeae* BEG12 and AZ225C enhanced alfalfa yield, nutrients, and fatty acid content. Similar findings were reported by Thioye et al. [47] where the growth-promoting effects of R. irregularis IR27 on Ziziphus mauritiana Lam continued 18 months after planting. However, Farmer et al. [51] and Alguacil et al. [49] could not correlate the persistence of inoculants with plant growth, even though inoculation increased plant growth in both studies.

Moreover, most studies reported a decline in the abundance of the persistent strains over time due to competition and suppression by native communities. For example, *R. irregularis* MUCL 41833 DNA was detected at a concentration 100 to 1000 times lower than native *R. irregularis* strains in three potato cultivars grown in the field in Belgium [68]. In a recent trial, the levels of the AMF abundance of inoculates introduced in four Canadian agricultural fields varied between 3 and 27 months [41]. *Rhizophagus irregularis* IR27 represented 11 to 15% of the *Rhizophagus* genus in *Ziziphus mauritiana* roots after 13 and 18 months [69]. A low abundance of introduced AMF compared to native strains indicates that exotic AMF can co-exist with native strains without posing a negative threat to local biodiversity. Lastly, the results obtained by Kokkoris et al., 2019 [46] also demonstrated that the establishment and persistence of AMF was site-dependent and not related to crop management. 

## 4. What Determines the Establishment Success and Survival of an Inoculum?

It is difficult to predict whether an introduced AMF will establish or fail; however, it is necessary to examine AMF root colonization more closely, which involves time-limited and host-independent pre-symbiotic and symbiotic phases occurring inside the root cortical cells. In a 2015 study, Bonfante and Desirò [70] indicated that there is no one single factor that can predict the post-application performance of AMF. Nevertheless, the survival of introduced AMF relies on inherent competitive traits, as well as the taxa, and the diversity of both the introduced AMF and the indigenous community [71]. Introduced species must be compatible both with the prevailing physical conditions of the soil and with the plant genotype [72,73,74]. Therefore, the factors affecting the performance of introduced inoculants in the field can be summarized as follows: the quality and type of the introduced strain, the local biotic and abiotic conditions of the target site, and priority advantage and propagule pressure (Figure 3).

### 4.1. Quality, Formulation, and Type of Inoculants

The success of inoculation depends on the inherent properties of inoculants such as the propagule type (i.e., spore, roots, hyphae, or mixture), the quantity formulation, the germination requirement and viability of the inoculant, the dose, and the frequency of inoculation [42,75]. Commercial inoculants lacking viable propagules will fail to colonize plant roots or compete with native AMF species. Some manufacturers’ claims about propagule compositions can be erroneous, as shown in Berruti et al. (2017) [53], where *Funneliformis* and *Septoglomus* were not detected in a commercial product, contrary to the manufacturer’s claims. Similarly, in a commercial AMF inoculum, *Funneliformis mosseae* BEG167 was confirmed, but only in low amounts compared to the contaminating *G. eutenicatum* [51]. Other products might have lost viability due to prolonged shelf-life, formulation, or inappropriate storage conditions, as observed by Salomon et al. (2022) [32].

The compatibility of the AMF strain with the target crop can also affect the establishment and survival of inoculants after inoculation. For example, Imperiali et al. (2017) [76] reported that *R. irregularis* SAF22 and another *R. irregularis* from a commercial inoculum were detected in roots, while *F. mosseae* SAF12, and *C. claroideum* SAF12 did not establish in wheat roots; these results are in line with the findings of Pellegrino et al. (2011) and Framer et al. (2007) [48,51]. Moreover, the appropriate dosage of inoculum must be considered as this is pertinent, not only for establishment success, but also to maintain an optimum AMF level in plant roots that does not offset plant symbiotic gain. For example, doses containing 50 to 100 spores of *G. intraradices* produced optimum root colonization that enhanced the targeted plant traits, whereas higher doses consisting of 200 to 400 spores resulted in higher root colonization that negatively impacted the plant traits in sugarcane (*Saccharum* spp. cv Mex 69–290) [77].

### 4.2. Priority advantage and Frequency of Application

To promote an inoculant’s chances of successful establishment in soil with a highly diverse native AMF community, introduced strains must have an inherent competitive advantage, strong mutualistic qualities, or a high propagule fostered via pre-inoculation, a high dosage, or repeated inoculation [40,78]. Pre-inoculations are usually carried out on seeds or root stocks before cultivation to forestall priority advantage, since mycorrhization operates on a first come, first served basis [78,79]. However, priority advantage does not always translate to greater inoculant establishment, as observed in vines [66]. Similarly, an increased frequency of inoculation can generate propagule pressure, helping propagules move closer to the roots or seeds (i.e., seed coating) and favoring competitiveness [74,80], but this is often not feasible due to technical difficulties, and is not always effective [46,60,61]. High propagule pressure may even decrease the benefits of symbiosis [81] or promote the proliferation of invasive species [82]. 

### 4.3. Soil Abiotic Conditions

The physical properties of the soil are also crucial for the survival and functioning of microbial inoculants [83]. The sources and quantity of P in the soil can affect the diversity of microbiome associating with plant roots, and can affect root colonization and the subsequent performance of the AMF inoculant [84]. Long-term fertilization with rock phosphate enhanced AMF association symbiosis in maize compared to triple superphosphate [85]. On the contrary, studies have shown that AMF can establish symbiosis with hosts even in the presence of high amounts of P. For example, the establishment of *R. irregularis* SAF22 in eight Swiss farmers’ fields positively correlated with total soil P and organic carbon [56,75].

AMF diversity and its co-occurrence network has been shown to decrease with long-term fertilization management; however, changes in the rhizosphere usually differ from the root endosphere. For example, AMF diversity decreased in a wheat rhizosphere but increased in the endosphere relative to the control in response to 35 years of NPK application [86]. Furthermore, AMF species respond differently to fertilization. For example, Ma et al. [86] also demonstrated that *Glomeraceae* were most dominant in both the rhizosphere and endosphere of wheat under long-term non-fertilization, whereas *Claroideoglomeraceae* and *Paraglomeraceae* were predominant in both biotopes under long-term fertilization. Moreover, the abundance of *Glomeraceae* in the endosphere and rhizosphere correlated negatively with the total and available P, whereas it correlated positively with the C/P ratio, indicating that soil organic carbon can promote AMF symbiosis. On the contrary, the abundance of *Paraglomus* correlated strongly with soil nutrient status; this suggests that *Paraglomeracea* can establish more successfully under high-input agricultural systems, while *Glomeracea* might be more successful in low-input agriculture.

Other studies indicate that drought can hinder the establishment and survival of AMF [87,88], while soil pH also influence the germination, growth, and distribution of AMF [89]. Negative impacts of acidic soil on AMF performance have been documented, whereas liming and pasteurization improved AMF performance [90,91].

### 4.4. Soil Biotic Conditions

Soil with a low diversity of the native AMF community or reduced niche overlap, especially arid or semi-arid soil, has been shown to promote inoculant establishment and persistence for many years [92]. Conversely, a high diversity of indigenous AMF communities can reduce or eliminate introduced inoculants in the long term, even after successful root colonization. Bender et al. [56] reported that the abundance of established inoculum was negatively correlated with that of native AMF species. Sato et al. [93] also demonstrated that the indigenous AMF community affected the establishment of introduced species, which had an abundance of 48.3% of all the AMF species, compared to fumigated soil, where it accounted for two-thirds (89.6%).

In addition to diversity, the species composition of the indigenous community can also affect the performance of introduced inoculants. For example, *R. irregularis* mostly survived in soil where conspecific strains were not part of the indigenous communities; meanwhile, survival was inconsistent in soil with a highly diverse and abundant indigenous AMF community due to greater competition, reducing survival to one or two years in some fields [41]. Bender et al. (2019) [56] also suggested that the presence of genetically related indigenous species tends to have a negative impact on the success of inoculants; sites with a high proportion of *F. mosseae*, *F. caledonius*, and *Paraglomus brasilianum* had negative mycorrhizal growth rates compared to sites with more divergent members, such as *Diversispora* spp., which had positive mycorrhizal growth rates. 

## 5. Effect of Inoculation on the Structure of Indigenous AMF Communities

The enhancement of soil microbial diversity leading to better soil health and greater plant performance is the main objective of inoculation. However, the effect of the introduced strains on local microbiota can be unpredictable due to the complexity of the rhizosphere [94]. AMF inoculant can change the community dynamics of indigenous species in many ways including suppression, stimulation, and exclusion (Figure 3). Unintended effects of inoculation may include loss of biodiversity, the promotion of plant diseases, and plant invasion that can result in economic losses. Given the growing concern about the risks that commercial inoculants pose to the biodiversity, it is imperative to monitor the consequences of inoculation on native species; however, only a few studies have examined commercial inoculants in the field (Appendix A). The development of highly sensitive markers and high-throughput sequencing is providing the scientific community with the opportunity to evaluate species richness and the diversity of native microbiota pre- and post-inoculation. Therefore, the major findings for the literature are highlighted below.

**Figure 3 microorganisms-10-01897-f003:**
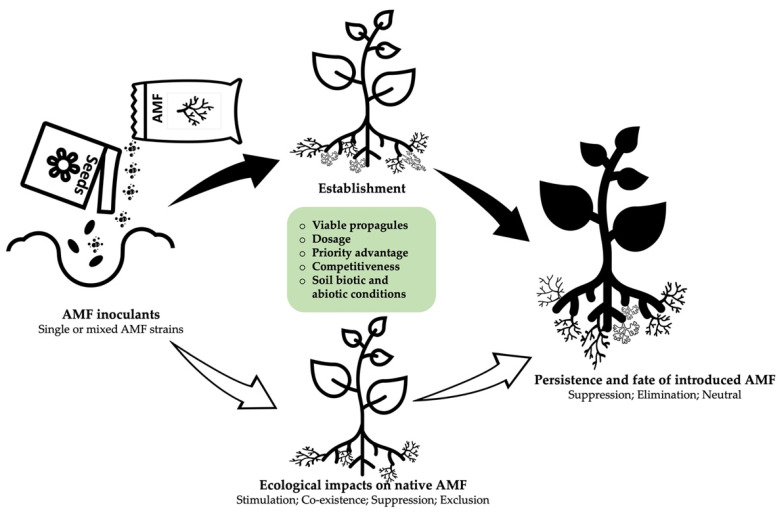
Overview of ecological impacts of AMF inoculants and driving factors.

### 5.1. The Impacts of Inoculation on Indigenous Communities Are Context-Specific

The introduction of AMF inoculants can induce neutral [47,52,54,57,63,95], negative [56,60,67], positive [53], or mixed [40,41,55] effects on the abundance and diversity of indigenous taxa (Table 2). These impacts can sometimes evade detection due to unfavorable conditions such as drought [87,88], as shown by Symanczik et al. [88] that the impacts of AMF on indigenous communities were only notable under wet conditions. In one case, inoculation induced a shift in the structure of resident communities [53] despite failure to colonize plant root. The alteration of native communities may result from microbial agents such as *Trichoderma* sp. and *Beauveria* sp., which were detected in the inoculum [53].

### 5.2. Inoculation mostly results into Shift in Structure, Rather Than in Composition

Most studies indicated that inoculation often influence the structure (relative abundance) of endogenous taxa rather than loss of species figure, especially in roots (Figure 2). The exclusion of certain taxa was however reported in a study [41], where *Rhizoglomus* and *Archaeospora* were not detected in the inoculated soil three years after inoculation. It is difficult to conclude, based on the above study, whether the exclusion of or failure to detect such AMF taxa resulted from inoculation alone, given that similar effects were observed in one control site where *Diversispora* only appeared during wheat rotation in the three cropping seasons, indicating that there may be other confounding variables (i.e., time of sampling) that could bias the detection of AMF taxa. 

The order of arrival and priority advantage are also crucial to determining the exclusion or inclusion of AMF in host roots [42,74]. If they have a minimum head start that confers a root colonization advantage [78], introduced strains can suppress and reduce the abundance of native species colonizing roots—although the frequency of this phenomenon differs drastically depending on the introduced AMF species or isolate. Pre-inoculation with *Septoglomus deserticola* (syn. *Glomus deserticola*) and *Claroideoglomus claroideum* syn. (*Glomus claroideum)* restricted other AMF taxa, while *Gigaspora margarita* and *Gigaspora gigantea* had no influence on indigenous AMF taxa [40]; other studies [47,66], however, found no effects or priority advantage. Regardless of priority advantage, the abundance of the introduced AMF diminishes over time, due to either competition with indigenous species or the lack of strong mutualism traits [41,56,67,90]; this is because plants allocate more photosynthates to beneficial symbionts, regardless of competitiveness [96]. The dormancy and reproductive periods of AMF species differ, especially when recovering from disturbance or seasonal variation [23,97]. Another possibility is that the excluded AMF taxa were dormant and not actively colonizing roots when the samples were taken; if this is the case, it will be necessary to consider both roots and soil in future investigations into the outcomes of inoculation, especially in the field.

### 5.3. Niche Availability Influences Fate of Both Introduced and Indigenous AMF Communities

Niche competition and the composition of the indigenous community are important ecological factors that determine the fate of the introduced inoculum. The introduced AMF can be suppressed by indigenous strains, or vice versa, if there is niche overlap between the communities. Thus, the AMF inoculum may stimulate certain AMFs from the indigenous community, while suppressing or excluding others from plant roots [41,55]. However, in the sites characterized by higher diversity of indigenous AMF species, inoculation can stimulate the abundance of AMF species in roots, without necessarily displacing native species [46,98]. Conversely, introduced strains may be eliminated in the presence of local ecotypes that have already adapted to the conditions of the site, such as agricultural disturbances and contamination, or in the presence of native AMF species that occupy the same niche [41,60,62]. Competition between AMF species can also occur between two similar strains inoculated together, as shown by Alguacil et al. (2011) [49], where root colonization by *G. intraradices* (syn, *Rhizophagus intraradices*) was decreased by 70% in the presence of *Glomus* sp.

### 5.4. Alteration in the Functions of Indigenous AMF Communities Are Scarcely Reported

Almost all the studies that investigated AMF either focus on plant performance or the effect on community structure or composition, how observed alteration in the community structure of indigenous AMF leads to shift in functional dynamics are largely unknown. Several studies have reported a positive response in plant traits (biomass and nutrient uptake) to AMF inoculation, but often fail to correlate it with mycorrhization intensity [19,51,53,99], suggesting that other mechanisms might be involved in the observed plant response. Furthermore, AMF species perform many ecological services, such as the promotion of soil aggregation through the secretion of glomalin-related protein, nutrient cycling, soil organic matter mineralization, etc. [34]. Therefore, an inclusive approach to studying the post-inoculation consequences of AMF introduction must account for all possible outcomes, including changes in plant fitness, carbon dynamics between the plant and AMF, carbon sequestration, metabolic activities, community structure, community composition, and the functions of native communities.

## 6. Monitoring Survival and Ecological Consequences of AMF Inoculants in the Field

Keeping track of introduced isolates among the large pool of indigenous AMF communities after inoculation is a complex task that requires the development of highly sensitive molecular markers that can identify AMF species at an isolate level. Absolute quantitative PCR is among the most accurate and reliable quantification methods that have been successfully utilized to trace AMF inoculants [100,101,102]. The establishment of introduced inoculants in roots can be assessed by comparing root colonization parameters and gene copy numbers, determined via absolute qPCR of the inoculated strains in control and in treatment plots [56]. The microscopic methods commonly used to assess inoculation success, by comparing the percentage root fragment colonized by AMF in the inoculated trial with a non-inoculated control, does not distinguish between species co-colonizing the same roots; thus, they cannot be relied upon to track the persistence of the introduced strains and disturbance in the indigenous community. Therefore, specific molecular tools are crucial to differentiating introduced strains from indigenous AMF communities when they are co-colonizing the same crop roots.

Metabarcoding studies monitoring AMF in the soil and roots usually target four nuclear ribosomal DNA loci; the partial small subunit (SSU), the large subunit (LSU), 5.8S rRNA genes, and the internal transcribed spacers (ITS), are the regions that are usually targeted with primers to differentiate AMF at the family and species levels [68,103]. Primers targeting one out of the four nuclear single rDNA loci can detect the presence or absence of specific fungi [51,95]; however, multiple copies of locus per genome, differences in the copy number among isolates, and genetic variations among these loci often generate low recovery of Glomeromycota sequences, especially when two isolates of a single AMF species are involved. Amplifying both the LSU and SSU regions together with ITS is a more sensitive alternative [95,104]. Thus, using two primers (NS31 and LSUGlom1) to amplify the 3′ end of the SSU and the 5′ end of LSU rRNA, together with the ITS region, Pellegrino et al. (2012) [67] traced an isolate of *Funneliformis mosseae* from an inoculum, using the polymerase chain reaction–terminal restriction fragment length polymorphism (PCR-(T)-RFPL) method. Pellegrino et al. [45] also targeted the SSU, ITS, and LSU regions to resolve both local and exotic *F. mosseae* and *R. irregularis* AMF strains. 

The partial SSU-ITS-LSU fragment is 1.5-kb long and cannot be directly sequenced by the second-generation high-throughput sequencing platforms, such as 454 pyrosequencing (fragment length approximately 800 bp) or MiSeq (approximately 500 bp); however, they can be sequenced by third-generation long-read sequencing technologies such as single molecular real times (SMRT) analysis provided by PacBio, which can read DNA sequences of more than 20 Kb, although with lower sequence quality than MiSeq [105]. Interestingly, Kolarikov et al. (2021) [106] successfully amplified the entire operon spanning the trio of SSU, ITS, and LSU in a two-step nested PCR using different primer combinations, i.e., AML1/LSUmAr and NS31/LSUmAr for the initial PCR and NS31_Glo3/LSUmBr for the second PCR. The 2.5 kb-long read was later sequenced using the single molecular real times (SMRT) analysis on the PacBio platform.

Intraspecific markers based on large subunits of a mitochondrial (mt) rDNA gene (*rnl* gene) are more reliable than each of the nuclear rDNA loci due to the lack of polymorphism in certain conserved domains and substantial variation among isolates, which enable distinctions to be made between haplotypes [96,107,108]. One technique involves the amplification of a region located in the *cox3-rnl* intergene of mtDNA, which harbors numerous mobile elements that have high sequence diversity; these include plasmid-related DNA polymerase genes (*dpo*), homing endonuclease genes, and small inverted repeats which are useful targets for the development of AMF strain-specific markers [109]. This technique has been successfully employed to distinguish between isolates of *Rhizophagus irregularis*, due to its insertion as a single-copy sequence and the absence of a nuclear type [100,101,102]. Badri et al. (2016) [102] employed this approach to develop a TaqMan-based qPCR method to quantify the spores of a commercial inoculant containing *R. irregularis* DAOM-197198 with high robustness and sensitivity. However, the major drawback of mtDNA-based markers is that they are only available for *R. irrgularisis* [101], and could be hindered by heteroplasmy produced by anastomosis; this may occur between compatible AMF isolates to form a hybrid consisting of both parents’ mitochondria, although it has been shown to be transient [110]. The occurrence of a hybrid progeny may affect the accuracy and interpretability of biodiversity indices, as such progenies can also form symbioses with plant roots [37].

Regardless of the method employed, there will always be a trade-off in terms of the cost, length of the sequence, and quality of the read. Notwithstanding, the various techniques must be standardized and harmonized to enable comparisons among studies and to eliminate the discrepancies that can stem from methodological biases towards certain AMF families [95]. 

## 7. Conclusions 

The application of bioinoculants remains an integral part of sustainable solutions to reduce the environmental footprints of conventional agriculture. However, it is necessary to address certain critical issues, such as the establishment and persistence of inoculants, as well as their short- and long-term impacts on indigenous communities; further research into these issues will inform management practices and regulations to avoid unintended consequences, such as the loss of native species and ecosystem functionality. The present review examines the fate of AMF inoculant intervention in the field from the perspective of establishment, survival, and whether they impact local communities, relying on the information available from field-based studies. We found that both the establishment and persistence of AMF inoculants are context-specific, and are affected by factors including inoculant quality, application practices, and local conditions. Some studies reported that the success of commercial inoculants tends to be more variable compared to laboratory strains, perhaps due to low-quality propagules or a loss of quality due to long-term storage conditions or erroneous claims by manufactures. After successful establishment, introduced AMF can survive for many years effects on crops are unpredictable. Furthermore, AMF inoculants may shift the community structure of native community assemblages by increasing or decreasing the relative abundance, while cases of complete exclusion are relatively scarce. 

## 8. Future Directions

The persistence, long-term performance, and the impacts of introduced AMF on indigenous species are influenced by both intrinsic (e.g., inoculant quality and viability) and extrinsic factors (e.g., soil and climatic conditions), which must be factored in when selecting AMF strains for commercial production. The intrinsic factors i.e., inoculant viability can be addressed perfectly by adopting the regulatory framework proposed by Salomon et al. [37] which can help forestall the proliferation of poor-quality products and win consumer confidence. However, continuous effort must be taken to monitor inoculant performance in the field under different edaphic and climatic conditions. Such studies will inform further transparency on product specifications and recommendations, as well policies towards the management of micro- and macro-biodiversity.

Moreover, the abundance and richness of indigenous AMF communities often correlates with soil physical properties such as soil organic matter (SOM), nitrogen, and precipitation [89,111,112,113,114], that can serve as clues for soil mycorrhizal potential. Such parameters can be modelled to predict inoculation success and post-inoculation effects on local microbiota. In addition, the distribution and abundance of AMF differ between the rhizosphere and root endosphere; while higher-endospheric populations can indicate greater establishment and persistence, it does not necessarily translate to greater plant benefits. On the contrary, lower AMF abundance in the endosphere versus higher extraradical abundance may indicate greater symbiotic advantage for plants [86]. Future studies need to investigate whether the abundance of AMF in the endosphere or rhizosphere is more important to plant response. However, in the meantime, samples from both biotopes, i.e., soil and root.

Are consortia the ideal inoculant to improve inoculant AMF establishment and persistence in the field, and to increase microbial diversity? AMF inoculants life cycle involves many stages including capture and refinement, production, establishment, persistence and function, and downstream impacts thus, beneficial traits in one context may be detrimental in another, necessitating a trade-off among these traits [36]. Selecting multiple AMF families having contrasting traits that favor diverse conditions can mitigate these compromises and help improve the chances of success in the field. A mixture of foreign inoculants consisting of *F. mosseae* and *R. irregularis* had greater persistence in alfalfa root than the respective single strain and local mixture, while also exerting the greatest impact on plant yield, nitrogen and phosphorus content, and fatty acids [45]. Merely increasing AMF diversity may not produce additional benefits [26]; however, the selected AMF should include distant family, which is likely to increase the possibility of finding complementary traits, as shown by Parhar et al. [115], who reported that consortia consisting of distant AMF families conferred greater additive effects than closely related ones. That is, the combination of more distant species, i.e., *R. fasciculatus* and *Gigaspora* sp., led to a greater effect on pea yield (50%) than *F. Mosseae* and *R. intraradices* (40%). Therefore, future studies should focus on screening and identifying more compatible phylogenetically distant AMF strains to develop commercial AMF inoculants that does not only productive in greenhouse but also successful under natural environments.

## Figures and Tables

**Figure 1 microorganisms-10-01897-f001:**
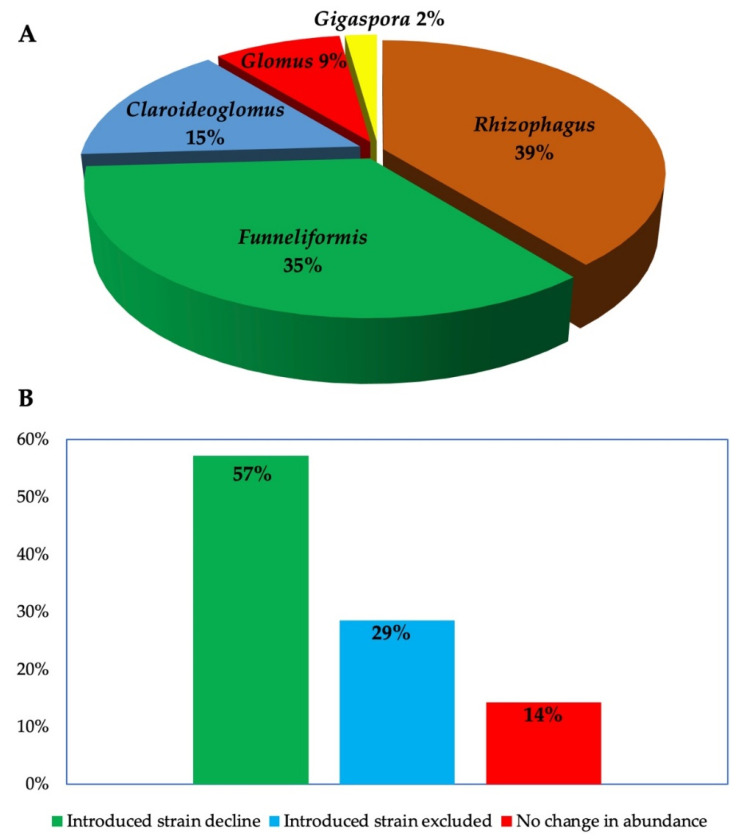
(**A**) AMF genera whose ecological effects were investigated in the field (**B**) Time-dependent changes in the abundance of introduced AMF according to results from six sites where the abundance of introduced AMF was monitored using metabarcoding twice or more. Introduced AMF: decreased in abundance compared to previous sampling; excluded (not detected in the sample) or abundance remained the same.

**Figure 2 microorganisms-10-01897-f002:**
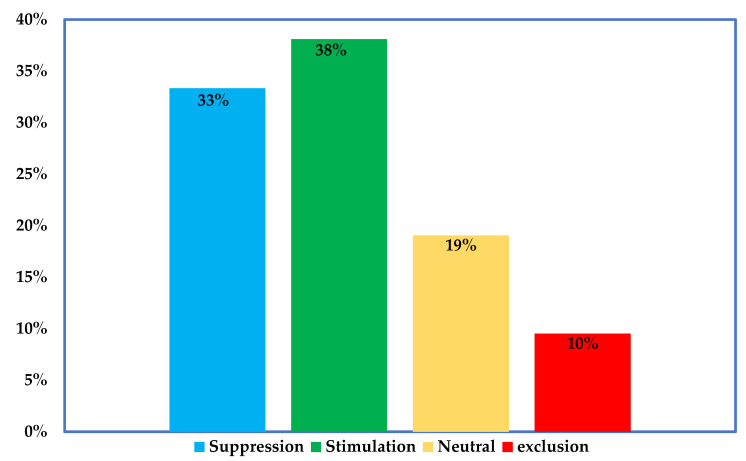
Impacts of inoculated strains on indigenous AMF under field condition. The figure was produced by case-by-case analysis of inoculants impacts as reported in the studies listed in Table 2.

**Table 1 microorganisms-10-01897-t001:** Persistence of introduced AMF under field conditions. Criteria for selection are experiment in field and use of molecular tools to trace or monitor introduced AMF.

AMF Name, Isolate (Brand, Manufacturer)	Location of Trial	Soil Type	Climate	Crop	Persistence Tracking Period	Change in Abundance of Introduced AMF over Time	References
*Funneliformis mossaeae* BEG12 and AZ225C, *Rhizophagus irregularis* BEG141	Manciano (Grosseto) Italy	Haplic Calcisol or Inceptisol	Humid Mediterranean	*Medicago sativa* L.	2 years	*F. mosseae* AZ225C was most persistent, followed by BEG12 and BEG141. Abundance of local *F. mosseae* cluster was reduced, while the abundance of native *R. irregularis* increased.	[45]
*Rhizophagus Irregularis*, DAOM 197198 (Myke Pro, Premier Tech)	Canada (Swift Current)	Brown Chernozem	Semi-arid	Pea-wheat	2 years, 5 months	Inoculant decreased from 27% in first season to 15% in years two and three.	[41]
	Canada (Outlook)	Dark Brown Chernozem	Semi-arid		2 years, 5 months	Introduced AMF decreased from 17% in year one to 4% in years two and three.	
	Canada (Scott)	Chernozem	Sub-humid		3 months	Relative abundance of introduced AMF was 33% but detection failed in subsequent years.	
	Canada (Melfort)	Black Chernozem	Sub-humid		1 year, 3 months	AMF detected in year one with 10% abundance and 4% in year two but declined completely in year three.	
*Rhizophagus irregulare*, DAOM 197198 (Myke Pro GR, Premier Tech)	Swift Current, Saskatchewan	_	_	Lentil *Lens culinaris*, *Linum usitatissimum* (Flax var. Bethune) rotation inoculation year 1 only	_	Inoculation did not affect abundance of target AMF in roots.	[46]
	Beaverlodge, Alberta	_	_	*Pisum sativum*, *Linum usitatissimum* (Flax var. Bethune)	_	Inoculation did not affect abundance of target AMF in roots.	
	Melfort, Saskatchewan	_	_		_	Abundance of introduced isolate was less in inoculated site than control	
	STR Saskatchewan	_	_		2 years	Abundance of introduced isolate was higher in inoculated plots in year one, but did not differ in year two.	
*Rhizophagus Irregularis* IR27	Senegal	Tropical ferruginous	Semi-arid	Jujube (*Ziziphus mauritiana* lam., Tasset and Gola)	1 years, 5 months	Abundance of *R. irregularis* was low (15% after 18 months).	[47]
*Funneliformis Mosseae* AZ225C and IMA1	Pisa, Italy	Sandy loam	Mediterranea climate	*M. sativa*	2 years	2 years, relative proportion of *F. mosseae* decreased in favor of native species, abundance of isolate AZ225C dropped from 100% to 16.3%, while isolate IMA1 survived only three months.	[48]
*Glomus* sp., *G. intraradices* and a mixture of both	Vicente Banes, Molina de Segura, Southeastern Spain)	Typic Torriorthent (silty clay)	Semi-arid Mediteeranean climate	*O. europaea*	1 year, 2 months	Abundance varied by AMF species: *Glomus* sp.: 20%, *G. intraradices*: 48.2%, and mixed (*G. intraradices*: 14%, *Glomus* sp.: 39.7%).	[49]
*G. intraradices* IMA6 and *F. mosseae* AZ 225C, and mixture of both	Italy		Mediterranean climate	Artichoke (*Cynara cardunculus* L. var. scolymus F.)	3 months	Inoculation increased the abundance of *Glomus* OTUs in inoculated plants compared to control.	[12]
*Gigaspora margarita* CK (cerakinkong, Central glass Co., Tokyo, Japan)	Mizunashi River, Mt Fugendake, Nagasaki Prefecture, Japan	Reforested soil		*Eragrosis curvula* (weeping love grass) and Miscanthus sinensis (Japanese Pampas grass)	4 years	*G. margariata* isolate CK was detected in rhizosphere and root of *E. curvula*.	[50]
*Funneliformis (syn. Glomus) mosseae* BEG12, 167 *G. intraradices* BEG 141 (IBG), *G. eutenicatum* BEG 168, (Endol, Biorize)	Daxing, Hebei Province, China	-		Sweet potato, (*Ipomoea batatas* L.)	3 months	Contaminating AMF, *G eutenicatum* had longer persistence than *G. mosseae*, the establishment of which was not successful in year 2; similarly, low amounts of *F. mosseae* BEG 167 were detected compared to contaminating *G. eutenicatum.*	[51]
*R. irregularis* GEG140, *F. mosseae* BEG95, *C. claroideum* BEG96, (Symbiom)	Coal mine spoil bank, Mekur, North Bohemia, Czech Republic	_	_	*Phalaris arundinacea*	3 years	Introduced AMF persisted and co-existed with native strains.	[52]

**Table 2 microorganisms-10-01897-t002:** Impact of introduced AMF on indigenous community. Criteria for selection were experiment in field and use of molecular tools to trace or monitor AMF communities.

AMF and Product Name	Molecular Method and Target Region	Crop	Location	Impacts on Native Community	References
*Funneliformis mossaeae* BEG12 and AZ225C, *Rhizophagus irregularis* BEG141	SSU-ITS-LSU (8SrDNA) sequencing	*Medicago sativa* L.	Manciano (Grosseto) Italy	Local *F. mosseae* clusters was suppressed while native *R. irregularis* stimulated	[45]
*R. irregularis* DAOM197198, (Myke Pro)	SSU-ITS-LSU (18S rDNA) 454 Pyrosequencing	Pea–wheat rotation	Canada (Swift Current)	Indigenous *Claroideglomus* was suppressed in third season.	[41]
			Canada (Outlook)	Abundance of *Glomus* and *Funneliformis* was decreased in year one, while *Claroideglomus*, *Paraglomus*, *Archaeospora*, and *Diversispora* were increased. *Rhozophagus* was excluded in third year.	
			Canada (Scott)	Indigenous *Claroideoglomus* and *Paraglomus* were stimulated, while *Funneliformis* decreased	
			Canada (Melfort)	*Glomus*, *Funneliformis* suppressed while *Claoroideoglomus* and *Paraglomus* stimulated. *Rhizoglomus* and *Archaespora* were excluded over three cropping seasons.	
*R. irregularis* IR27, lab-made inoculum propagated in greenhouse trap culture	LSU (18S rDNA) Illumina MiSeq	Jujube (*Z. mauritiana*	Senegal	Inoculation decreased *Rhizophagus*/*Glomus* ratio.	[47]
*F. coronatum* GO01, GU53, *F. Caledonium* GM24, *R. intraradices* GB6 and GG32, *F. mosseae* GP11 and GC11, and *Septoglomus viscosum* (GC41)	SSU (18S rDNA) 454 pyrosequencing	Fodder maize (*Zea mays* L. var. ‘Kalumet’	Carmagnola, Italy	Inoculation induced an increase in alpha-diversity indices in roots by reducing species dominance.	[53]
*R. intraradices*, *F. mosseae*, and mixture of both (MycAgro Lab)	Illumina MiSeq (ITS2)	Saffron (*Crocus sativus* L.)	Saint Christophe, Italy (Morgex, Aosta Valley, Italy)	Inoculation did not impact field fungal communities; results differed by year of sampling and field.	[54]
*Glomus* sp.	Illumina MiSeq (18S rDNA)	Welsh onion cv. Motokura	Okasaki, Ayabe, Tsugaru, Japan	OTUs related to introduced AMF were decreased, while those belonging to distant taxa such as *Gigaspora*, and *Acaulospora* were consistently enriched.	[55]
*R. irregularis* GEG140, *F. mosseae* BEG95, *C. claroideum* BEG96 (Symbiom)	PCR-RFLP (25S rDNA)	*Phalaris arundinacea*	Coal mine spoil bank, Mekur, North Bohemia, Czech Republic	AMF persisted for 3 years and co-existed with native haplotypes of same species.	[52]
*Rhizophagus irregularis* SAF 22 Blazk, Wubet, Renker and Buscot	SSU-ITS-LSU (18S rDNA) SMRT and qPCR	Swiss corn	Eight farmers’ fields	Establishment of inoculated strains correlated negatively with root colonization.	[56]
*R. irregularis* DAOM 197198 (Myke Pro Liquid, Myke Pro Soybean Liquid, Myke Pro PS3, Premier Tech)	ITS-SSU (18S rDNA) Illumina MiSeq	*Zea mays*, cultivar Elite 49A12 Cruiser Max Quattro,	St-Elzear Quebec	Inoculation did not affect abundance or community diversity.	[57]
		*Glycine max* cv. Pioneer, 90YO1,	Notre-Dame-du-Mont-Carmel, Quebec		
		Wheat (*T. aestivum*), cv. Touran	Sainte-Helene-de-Kamoraska, Quebec		
*Glomus* spp.	SSU (18S rDNA) PCR-RFLP	Spice pepper (*C. annuum* L. var. longum), cv. Szegedi and cv. Kalocsai		Inoculation affected structure of resident AMF community, but there was no remarkable effect on AMF species composition.	[58]
*G. intraradices* BEG140, *G. mosseae* BEG95, *G. etunicatum* BEG92, *G. claroideum* BEG96, *G. microaggregatum* BEG56, *G. geoposporum* BEG199 (Symbivit, Symbiom)	PCR-RFLP (18S rDNA)	*Capsicum annuum* L. var. longum and cv. Szegedi	Godollo, Hungary, Continental	Inoculation affected relative abundance of AMF ribotypes but did not influence composition.	[59]
*R. irregularis* DAOM 197198 (AGTIV)	Illumina MiSeq (ITS, mitochondrial rDNA)	Flax, lentil	Saskatchewan (Swift Current)	Single inoculation had no effect, but continuous AMF inoculation reduced Shannon diversity and Pielou’s evenness indices in flax rhizosphere in second rotation in Beaverlodge.	[60]
			Alberta (Beaverlodge)		
*R. irregularis* GD50	Illumina MiSeq (ITS)	Lentil–wheat,	Swift Current, Saskatchewan	No effect of inoculation in rotation phase 1, but AM altered fungal community structures of rhizosphere and root of flax grown in Swift Current in rotation phase 2.	[61]
		Pea–flax	Beaverlodge		
*R. irregularis* R-10	Illumina MiSeq (25S rDNA)	*Glycine max* (L.) *Merrill.* cv. *Fukuyutaka*	Kyushu, Okinawa Agricultural Research Center, Miyakonojo, Miyazaki Japan	Inoculation increased read abundance of inoculum *R. irregularis* (70%) compared to 30% in non-inoculated site, by competing for niche commonly distributed communities.	[62]
*R. irregularis* DAOM 19178 from four products: Myke Pro p-801, Myke Pro GR, Mycorise ASP, and Symplanta	454 pyrosequencing (25S rDNA)	*Solanum tuberosum* c.v. INIAP-FIpapa	Zamora Huayco Research station, Loja	No effect on indigenous AMF community. Introduced AMFs were outcompeted by indigenous *Acaulospora* sp.	[63]
			Santa Catalina Research Station, Ecuador		

## Data Availability

Not applicable.

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
