# Peer review of "The Potential Applications of Commercial Arbuscular Mycorrhizal Fungal Inoculants and Their Ecological Consequences"

_microorganisms, 2022, doi:10.3390/microorganisms10101897_

Round 1
Reviewer 1 Report
The manuscript is a review of the artciles dedicated to the investigation of the effects of commercial AMF inoculants in field conditions. Although the manuscript itself is well-constructed and easy to read, there are some problems, that make it difficult to judge the soundness of the work.
1) The databases, in which the search was conducted were not named, making it impossible to verify the conducted study
2) The found artciles were filtered according to the criteria described in the text, but the full list of found articles before filtering was not presented. It is thus impossible to judge, whether the article is indeed a thorough review without performing the same search and analysis for yourself.
Author Response
We thank the reviewer for providing helpful comments that improved our Manuscript.
Responses to the Reviewer's comments point by point:
- The databases, in which the search was conducted were not named, making it impossible to verify the conducted study
Response: The databases are Web of science and Scopus
- The found articles were filtered according to the criteria described in the text, but the full list of found articles before filtering was not presented. It is thus impossible to judge, whether the article is indeed a thorough review without performing the same search and analysis for yourself.
Response: List of all articles now available in the supplementary material as a list of references.
Reviewer 2 Report
Dear authors, I appreciate your well written review on “The Potential Applications of Commercial Arbuscular Mycorrhizal Fungal Inoculants and their Ecological Consequences”. The topic is relevant, the subject is very interesting, and the review is well organized. I highly recommended to publish this manuscript in “Microorganisms”. That being said, the manuscript has the potential to be accepted. However, there is still some minor issues need to be addressed before the paper could be accepted as follows:
Comments
Please consider changing the keywords list and use synonyms.
Line 34: This sentence is unclear. Give examples to environmental consequences and define 4R framework.
Line 57-59: Give some examples regarding fungal species competition, in some cases the mixed application can add more benefits.
Line 192-193: not clear
Line 93: Insufficient, some discussions are limited and can be enlarged in a separate section using recent and appropriate literature. For example, these are important to discuss due to the impact of AMF inoculation on microbial community in the soil which can add more benefits to soil fertility e.g.
https://doi.org/10.1186/s12870-021-02949-z
https://doi.org/10.3390/agriculture11030194
https://doi.org/10.3390/agronomy10030319
Lines 380-429: Be more concise. What is the novelty of this work?
I highly recommended to add “A Future Prospects” in a separate section to allow readers to express their thoughts on the future of this research.
The number of factors discussed in this review is relatively large, so that the key content is not easy to focus on. Therefore, a mechanism diagram or pattern diagram is suggested to add in the review to present the process outputs.
Kind Regards
Author Response
We thank the reviewer to taking the time to read and think about our Manuscript. We found the comments very helpful to make a major revision which includes a new Figure 3. We have changed the text of some sections to address the reviewer's comments.
Responses of the Reviewer's Comments point by point:
Please consider changing the keywords list and use synonyms.
Response: We changed keywords and synonym as suggested by the reviewer.
Line 34: This sentence is unclear. Give examples to environmental consequences and define 4R framework.
Response: Examples of environmental consequences including greenhouse gas emission, eutrophication, pollution of surface and underground water, global soil loss to salinity and compaction, loss of microbial diversity are now given in the article. Provided. 4R means means the right type and quantity of nutrient at right place and right time.
Line 57-59: Give some examples regarding fungal species competition, in some cases the mixed application can add more benefits.
Response: Benefit on AMF consortia either fungus-fungus or fungus-bacteria are stated. Reference was given where higher fungi diversity did not work.
Line 192-193: not clear
Line 93: Insufficient, some discussions are limited and can be enlarged in a separate section using recent and appropriate literature. For example, these are important to discuss due to the impact of AMF inoculation on microbial community in the soil which can add more benefits to soil fertility e.g.
https://doi.org/10.1186/s12870-021-02949-z
https://doi.org/10.3390/agriculture11030194
https://doi.org/10.3390/agronomy10030319
Response: Thanks for these references. The roles of AMF are mentioned with regards to plant and soil (lines 43 to 53) using information from the recommended references and others. The articles generally discussed impacts of inoculation on AMF community.
Lines 380-429: Be more concise. What is the novelty of this work?
I highly recommended to add “A Future Prospects” in a separate section to allow readers to express their thoughts on the future of this research.
The number of factors discussed in this review is relatively large, so that the key content is not easy to focus on. Therefore, a mechanism diagram or pattern diagram is suggested to add in the review to present the process outputs.
Response: Separate sections are added in conclusions and future direction. The conclusion states the novelty of the paper and summarizes the major findings. We added a new Figure 3 that shows an overview of mechanisms and interaction patterns as suggested by the reviewer.
Round 2
Reviewer 1 Report
The comments were adressed appropriately.